# Tissue-specific mitochondrial pathway remodeling linked to longevity in honeybee queens

Clément Joël Lucien Chevret[1,2], José Francisco Echegaray[1], Alexander Walton[3,4], Maryam Lo[1,2], Olav Rueppell[3], Hélène Lemieux[1,2]*

1 Faculty Saint-Jean, University of Alberta, Edmonton, Alberta, Canada, 2 Department of Medicine, University of Alberta, Edmonton, Alberta, Canada, 3 Department of Biological Sciences, University of Alberta, Edmonton, Alberta, Canada 4 Department of Biological Sciences, Concordia University of Edmonton, Edmonton, Alberta, Canada

* helene.lemieux@ualberta.ca

## Abstract

Mitochondrial metabolism plays a critical role in determining lifespan across animal taxa. In our study, we used the Western honeybee (*Apis mellifera*) as a model, capitalizing on the stark lifespan difference between queens, which often live more than two years, and summer workers, which survive only about 30 days, despite sharing the same genetic background. We investigated mitochondrial function in head tissue, thoracic muscle, and abdominal fat tissue of queens and workers, comparing early (7 days) and late adult stages (28–30 days in workers; 2 years in queens). No significant differences in mitochondrial flux control ratio for the NADH- Succinate- and glycerophosphate (Gp) pathways were found in thoracic muscles across castes or age groups. In head and abdominal fat tissues, early-life queens showed reduced reliance on NADH-linked pathways for maximal respiratory flux compared to workers. The decrease in the NADH-pathway was compensated by an increase in the Gp-pathway contribution. Queens exhibited reduced phosphorylation-pathway control over OXPHOS compared to workers, both in head tissue during early life and in abdominal fat tissue later in life. These findings reveal caste- and tissue-specific patterns of mitochondrial regulation that may contribute to dramatic lifespan divergence observed in eusocial insects. They suggest that early-life metabolic flexibility could play an important role in shaping life history evolution in *Apis mellifera*.

## 1. Introduction

Physiological aging in animals involves progressive, stochastic declines in both global and specific cellular functions, accompanied by an increased vulnerability to disease [1]. Over the past century, numerous theories have been proposed to explain the biological basis of aging (see reviews [1–5]), including mitochondrial

**Data availability statement:** https://doi.org/1010.5061/dryad.jdfn2z3q4.

**Funding:** This study was supported by the following grants: two Discovery Grants from the Natural Sciences and Engineering Research Council of Canada (RGPIN-2021-02924 to H. Lemieux and RGPIN-2022-03629 to O. Rueppell), and a grant from the U.S. Army Research Office (W911NF-22-1-0195 to O. Rueppell). The funders had no role in study design, data collection and analysis, decision to publish, or preparation of the manuscript.

**Competing interests:** The authors have declared that no competing interests exist.

dysfunction, genomic instability, telomere attrition, epigenetic alterations, loss of proteostasis, deregulated nutrient sensing, cellular senescence, stem cell exhaustion, and altered intercellular communication. While aging clearly arises from multiple interconnected processes, many of these pathways ultimately converge on mitochondria as central integrators of metabolic regulation and cellular homeostasis. One notable exception is telomerere attrition, which is not universal and does not apply in *Apis mellifera*, where telomerase activity remains high and telomeres remain stable throughout life [6].

Even subtle differences in mitochondrial protein abundance, gene expression, or functional efficiency can exert outsized effects on lifespan and health. This is because mitochondrial metabolism is not solely responsible for energy supply, it also shapes oxidative stress, cell signaling, and overall homeostasis. Consistent with this idea, comparative studies across taxa have repeatedly linked variation in mitochondrial performance with lifespan differences in mammals [7–10], insects [11–15], bivalves [16], and *C. elegans* [17–25]. This conserved link between mitochondrial function and aging underscores this organelle as a key hub in longevity regulation and in age-related pathologies such as cancer, cardiovascular disorders, and neurodegeneration—all conditions linked to mitochondrial dysregulation [26].

Among aging research models, social Hymenoptera provide a powerful system to investigate how metabolic regulation and developmental plasticity shape lifespan variation. Honeybees exhibit two levels of phenotypic flexibility. First, genetically identical individuals can develop into either queen or worker castes. In the Western honeybee (*Apis mellifera* L.), queens can live more than two years, whereas spring and summer workers typically survive only 3–6 weeks [27]. Second, within the worker caste, age-based division of labor (temporal polyethism) is associated with marked differences in physiological aging pace: young workers perform in-hive tasks such as nursing with relatively slow aging, while older workers transition to foraging outside the hive, a task associated with accelerated physiological decline [28,29]. In contrast, queens undertake energetically demanding mating flights early in life to mate with multiple males whose sperm will sire the queen's female offspring for the entirety of their life [30]. Following these flights, queens adopt a largely stationary lifestyle, only moving around the brood nest where each queen can lay up to 1,000–2,000 eggs daily [30], which imposes substantial metabolic demands related to reproduction. Caste differentiation is triggered by diet during larval development: queen-destined larvae are consistently provided with large amounts of proteinaceous royal jelly, while worker larvae receive a mix of worker jelly, honey, and pollen [30]. This nutritional divergence induces major epigenetic reprogramming and results in substantial transcriptome differences between castes (reviewed by Alhosin [31]). These profound intraspecific differences in lifespan, behavior and physiology are likely linked to finely tuned metabolic adaptations, especially mitochondrial function. However, we still lack a comprehensive understanding of the specific mitochondrial function differences across tissues, life stages, and castes in honeybees. It also remains unclear whether caste-specific metabolic profiles are established early through nutritional and epigenetic programming, shaped later by adult behavior and physiology, or influenced

by both. Understanding this complexity is crucial, as even subtle, tissue-specific differences in mitochondrial function may contribute to differential rates of physiological decline or promote lifespan in honeybees.

One limitation of previous honeybee research is the scarcity of comprehensive, tissue-specific analyses of mitochondrial function across castes with markedly different lifespans. While some studies have investigated specific aspects of mitochondrial physiology in isolated tissues, few have taken an integrative multi-tissue and multi-caste approach. The study by Corona et al. [32], which examined mitochondrial gene expression across the brain, thorax, and abdomen in both queens and workers, is one of the rare exception that spans multiple tissues and directly compare the two castes. They found that aging in queens, but not in workers, was associated with reduced expression of mitochondrial genes, including mitochondrial-encoded (cytochrome b, COX-I) and nuclear encoded (cytochrome cclock-1, mitochondrial translation initiation factor 2). Additionally, antioxidant gene expression differed only slightly between castes, leading the authors to suggest that antioxidant capacity plays a limited role in queen longevity, and if it does contribute, it might be restricted to the abdomen. To add to these valuable insights, Hsu et al. [33] investigated abdominal trophocytes and oenocytes in young and old queens and workers, revealing that energy utilization—measured by $NAD^+$ levels and the $NAD^+/NADH$ ratio—declined with age in workers and was consistently lower in workers than in queens. Other studies focusing exclusively on worker bees have revealed that mitochondrial function changes over time and time labor division in key tissues, including the brain [34,35], thoracic muscle [34,36], and abdominal fat bodies [34,37]. Collectively, these studies highlight important components of mitochondrial biology in honeybees, yet a comprehensive, tissue-specific characterization of pathway-level mitochondrial function across castes and life stages remains lacking.

The objective of our study was to compare workers and queens, and to examine early and late adult life stages within each caste, to identify tissue- and caste-specific modifications in mitochondrial oxidative phosphorylation (OXPHOS) pathways in the honeybees. We focused three key anatomically and functionally distinct regions: the head, the thoracic muscle, and the abdominal fat bodies. Each body part has distinct and essential roles in bee physiology and behavior. The head contains the brain, which governs complex behaviors such as navigation, communication, memory, learning, and decision-making, all highly dependent on mitochondrial energy. The thorax houses flight muscles, among the most metabolically active tissues, especially in foraging workers (reviewed by [38]). In queens, these muscles power mating flights early in life and swarming flights late in life, but remain largely inactive in between, serving primarily for vibrational communication. The abdomen (after removal of the alimentary tract, ovaries and spermathecal) contains predominantly the fat bodies, a multifunctional organ akin to the mammalian liver and adipose tissue, supporting energy storage, metabolic regulation, detoxification and immune responses [39]. Our functional measurement focused on specific key OXPHOS pathways feeding electrons into the electron transport system (ETS) in honeybees [36,38,40–43]: the NADH-linked pathway (via complex I), the succinate pathway (via complex II), and the glycerophosphate pathway (via mitochondrial glycerophosphate dehydrogenase, a key pathway in insect flight muscle metabolism [44]). We also quantified Complex IV activity, representing the electron transport system's final step, and citrate synthase activity (recognized as a strong marker of mitochondrial content [45]) to distinguish changes in respiratory capacity from differences in mitochondrial abundance. By characterizing these distinct OXPHOS-linked pathways and steps across anatomically and functionally distinct body regions, we aimed to gain a comprehensive understanding of how mitochondrial function varies across castes and life stages, and whether these differences align with the reduced physiological decline and extended lifespan characteristic of queens.

## 2. Methods

### 2.1 Bee colonies

Adult honeybees of different ages and castes were used. Workers (early-life, 7 days; late-life, 28–30 days) and early-life queens (7 days) were sourced from colonies at the Rueppell Lab Apiary on the North Campus of the University of Alberta (Edmonton, Alberta, Canada). Early queens (7 days old) were produced through standard emergency queen-rearing

methods: brood frames containing open worker larvae were placed into queenless nucleus colonies, which reared new queens from this brood. Queens were open-mated with local drones, and their mated status was confirmed upon dissection by assessing the presence of seminal fluid within the spermathecae. To obtain age-matched workers, brood frames containing pre-eclosion pupae were removed from four source colonies and incubated overnight at 33°C to ensure emergence within a 24-hours window. Newly emerged workers were gently brushed into plastic tubes, marked with enamel paint (Testor Corporation, Rockford, IL) on the thorax to signify age cohort and introduced into four healthy, queenright experimental colonies used for housing the marked bees. Early-life queens were marked similarly. Day-old workers from multiple source colonies were pooled in a single tub prior to introduction to minimize colony-source effects [46]. Late-life queens (2 years old) were not generated for this experiment, but were commercially reared and housed in colonies at the University of Lethbridge (Lethbridge, Alberta, Canada) since June 2022. These queens originated from the same commercial maternal lineage as the worker/early-queen stock. All experimental, age-matched bees, were descendants of unselected commercial stock representing mixed European lineages [47]. All experimental colonies were healthy, queenright and free-flying throughout the study. Marked worker cohorts were generated repeatedly between June and August 2024, using distinct paint colors to track age. Bees were sampled randomly from frame within the four experimental colonies at designated time points for mitochondrial function analysis. Final sample size were: workers at 1 week (early-life, $n = 8$) and 4 weeks (late-life, $n = 12$) and queens at 1 week (early-life, $n = 6$) and 109 weeks (late-life, $n = 4$). Group sizes reflect availability and biological constraints.

## 2.2 Mitochondrial function measurements

Bees were anesthetized in a falcon tube on ice for 5–10 minutes until all movement ceased. Each bee was transferred to a plastic plate. Using dissecting scissors, the head, was carefully separated. An incision was made along the dorsal abdomen. The alimentary tract (midgut and hindgut), ovaries and spermatheca were extracted by gently pulling on the stinger with forceps. The thorax and abdomen were then separated. Each body part was weighted using an analytical balance (Mettler Toledo XS205) and immediately placed in an individual glass petri dish on ice. Ice-cold Mitochondrial Respiration Medium 05 (MiR05; 0.5 mM EGTA, 3 mM $MgCl_2$-$6H_2O$, 60 mM K-lactobionate, 20 mM taurine, 10 mM $KH_2PO_4$, 20 mM HEPES, 110 mM sucrose, and 1 g·L$^{-1}$ BSA essentially fatty-acid free, pH 7.1; [48]) was added –500 μL for the head and abdomen, and 1 ml for the thorax. Tissues were cut into 1–3 mm fragments using a razor blade and transferred to an ice-cold Potter-Elvehjem tube. Petri dishes were rinsed three times with 500 μL of MiR05 (head and abdominal fat tissues) or 4 times with 1 ml (thoracic muscle). Final MiR05 volumes were adjusted to 2 mL for abdominal fat tissue (no additional volume), 10 mL for head tissue (8 mL added), and 30 mL for thoracic muscle (25 mL added). An overhead stirrer (Wheaton Instruments, Millville, NJ) was used for five passes at a speed of 1.0. These procedures were optimized based on preliminary assays to maximize respiration rates, preserve coupling, and maintain outer mitochondrial membrane integrity. A portion of each homogenate was stored at −80°C for citrate synthase (CS) activity assays, performed in replicate as previously described [49]. Enzymatic activity was expressed in international units (IU) per milligram of tissue, where one IU corresponds to the conversion of one μmol of substrate per minute.

Immediately after homogenate preparation (typically within 2 minutes), samples were introduced into the six chambers OROBOROS Oxygraph-2k (OROBOROS Instruments Inc., Innsbruck, Austria), and mitochondrial respiration was measured at 33°C. Chambers were pre-calibrated with 2 mL of MiR05. The protocol (Fig 1) assessed three respiratory states: leak state (non-phosphorylating, no ADP), coupled OXPHOS capacity (with saturating ADP), and ETS capacity (decoupled, via uncoupler titration) [50]. Pyruvate (5 mM), malate (2 mM), and glutamate (10 mM) were added to each chamber containing 1.900 μL MiR05 (thoracic muscle) or 1.800 μL (head and abdominal fat). Then, 100 μL of thoracic muscle homogenate and 200 μL of head/abdominal fat homogenate was added to reach 2 mL total volume (leak state). Substrates, uncoupler, and inhibitors were added sequentially: 2.5 mM ADP (OXPHOS via the NADH-pathway through Complex I), 10 μM cytochrome $c$ (outer mitochondrial membrane integrity), 10 mM succinate (combined NADH and Succinate

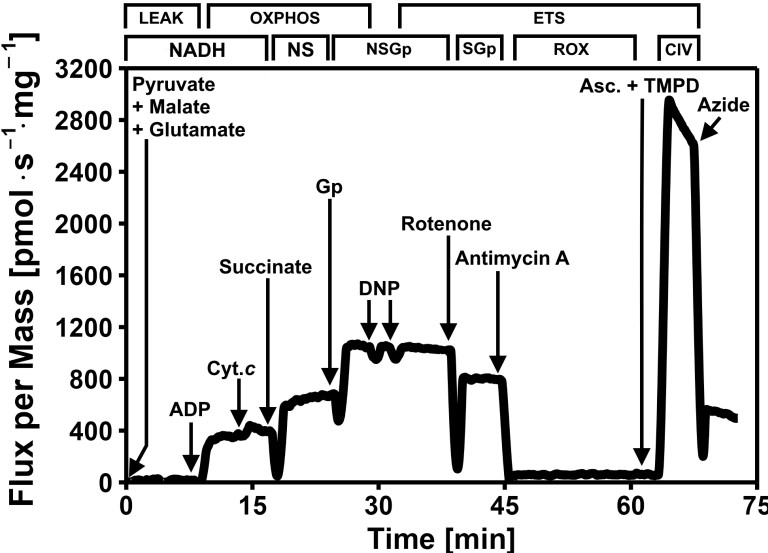

**Fig 1. Representative trace of high-resolution respirometry using a multiple substrate—uncoupler-inhibitor titration protocol in the thoracic muscle of worker honeybees (*Apis mellifera*).** The oxygen consumption (y-axis) per mg of tissue mass is represented as a function of time (x-axis). Arrows indicate the timing of titrations of substrates, uncoupler and inhibitors. The protocol includes the following steps: pyruvate+malate+glutamate added before the homogenate (NADH-pathway, LEAK state), ADP (NADH-pathway, OXPHOS capacity); cytochrome *c* (Cyt. *c*; integrity of outer mitochondrial membrane), succinate (combined NADH and succinate pathways, NS-pathway, OXPHOS capacity), glycerophosphate (combined NADH, succinate, and Gp pathways, NSGp-pathway, OXPHOS capacity), uncoupler (dinitrophenol, DNP) titration (NSGp-pathway, maximal ETS capacity), rotenone (inhibition of Complex I, SGp-pathway, ETS capacity), antimycin A (inhibition of complex III, residual oxygen consumption, ROX), ascorbate (Asc.) and TMPD (Complex IV activity) and sodium azide (inhibition of Complex IV, chemical background).

OXPHOS pathways through CI and CII; NS-pathway), 10 mM Gp (combined NSGp OXPHOS pathway), dinitrophenol (DNP) titration up to a concentration between 0 and 0.175 mol L$^{-1}$ (maximal ETS capacity; combined NSGp-pathway), 1 μM rotenone (ETS capacity; SGp-pathway), 2.5 μM antimycin A (non-mitochondrial residual oxygen consumption, ROX), 2 mM ascorbate and 0.5 mM N,N,N',N'-tetramethyl-p-phenylenediamine (TMPD) (complex IV activity), and 100 mM sodium azide (chemical background). The protocol was developed based on a key substrates relevant to bee metabolism [36,38,40–43], and refined through preliminary experiments. Data collection occurred from June 06 to Aug. 28, 2024, to avoid potential, albeit likely minor [41], effects of wintering stages on worker or queen metabolism.

To interpret Gp-pathway results, we also assessed OXPHOS coupling and respiratory capacity using fatty acid substrates. Although thoracic muscle reportedly cannot oxidize fatty acids [38,41,43], it remains unclear whether head and abdominal fat tissues can. We therefore tested mitochondrial fatty acid oxidation (FAO-pathway) in worker bees using two protocols: 0.1 mM malate, 10 μM palmitoylcarntine (protocol 1) or 0.2 mM octanoylcarnitine (protocol 2), 2.5 mM ADP, 10 μM cytochrome *c*, 2 mM malate, 5 mM pyruvate, 10 mM glutamate, 10 mM succinate, uncoupler titration (dinitrophenol, DNP, up to a concentration between 0 and 0.175 mol L$^{-1}$; FAONSGp-pathway), and 100 mM sodium azide.

High-resolution respirometry data were analyzed using Datlab 7.4 software (OROBOROS Instruments Inc.). Mitochondrial respiration was expressed as flux per mass (pmol O$_2$.s$^{-1}$.mg$^{-1}$ tissue, corrected for ROX, except CIV, which was corrected for chemical background). Alternatively, respiration was expressed as the flux control ratio (FCR; [50,51]), defined as the respiration of a specific pathway or complex (Yi) normalized for the rate Z, i.e., the maximal ETS capacity (NSGp- or FAONSGp-pathway).

$$FCRi = \frac{Yi}{Z}$$

FCRs, by comparing oxygen flux in different respiratory states to a common reference state, allow comparisons that are independent of mitochondrial content or assay conditions.

Outer mitochondrial membrane integrity was assessed using cytochrome *c* control efficiency, calculated as:

$$\frac{\text{OXPHOS flux with cytochrome } c \; - \; \text{OXPHOS flux without cytochrome } c}{\text{OXPHOS flux with cytochrome } c}$$

Cytochrome *c* control efficiency was low across most tissues: median of 0.01 for the head (range: 0.00–0.22), 0.06 for abdominal fat (0.00–0.32), and 0.12 for thoracic muscle (0.00–0.24). To avoid bias due to variation in outer membrane integrity, OXPHOS capacity comparisons across ages and castes used values after exogenous cytochrome *c* addition.

### 2.3 Statistical analysis

All statistical analyses were performed in R (v4.4.2, R Core Team, 2024) using RStudio (Posit Team, 2024; RRID:SCR_000432). Details are provided in the figure legends. For each tissue (head, thoracic muscle and abdominal fat) and pathway or step, normality and homogeneity of variance were tested using Shapiro-Wilk, Bartlett's and Levene's tests. If both criteria were met, a two-way ANOVA followed by Tukey's HSD test was performed. If only normality was met, Welch's ANOVA with Games-Howell post-hoc test was used. If the normality was not met, Kruskal-Wallis test with Dunn's test was used. Analyses used the *car* (RRID:SCR_022137), *DescTools* (https://doi.org/10.32614/CRAN.package.Desc-Tools), *rstatix* (RRID:SCR_021240), and *FSA* (https://doi.org/10.32614/CRAN.package.FSA) packages. Figures were generated using *ggplot2* (RRID:SCR_014601), *patchwork* (RRID:SCR_024826) and *cowplot* (RRID:SCR_018081). Statistical significance was set at $P \leq 0.05$.

## 3. Results

### 3.1 Tissue mass and mitochondrial content

Head mass was consistent across age and caste (Fig 2A). Queens had greater thorax and abdomen mass than workers in both early-life ($P \leq 0.001$ for thorax and abdomen) and late-life (thorax: $P \leq 0.001$; abdomen: $P \leq 0.05$; Fig 2B, 2C). In workers, thorax and abdomen masses remained unchanged with age (Fig 2B, 2C). In queens, thorax mass increased with age ($P \leq 0.01$; Fig 2B) while abdomen mass remained stable ($P = 0.286$ Fig 2C).

Mitochondrial content was assessed using CS activity (Fig 2D-2F), respiration rate through combined electron transport system pathways (ETS; Fig 2G-2I), and CIV activity (S1 Fig A-C), all normalized to tissue mass. These are commonly accepted markers of mitochondrial content, with CS activity being particularly well-established [45,52]. In head tissue (Fig 2D), CS increased with age in workers ($P = 0.007$) but not in queens ($P = 0.112$). As a result, old queens had significantly lower CS activity than old workers ($P < 0.001$), with no difference at younger ages ($P = 0.779$). ETS capacity showed a similar pattern in the head tissue (Fig 2G). In contrast, CIV activity per mg head tissue did not vary by caste or age (S1 Fig A). In thoracic muscle, all three mitochondrial markers showed no significant difference between caste or age (Figs 2E, 2H, S1 Fig B). In abdominal fat tissue, patterns were also consistent. Workers showed no age-related change ($P = 1.000$ for CS activity, ETS capacity, CIV activity, Figs 2F, 2I, S1 Fig C). Young queens had higher mitochondrial content than old queens, but this was only significant for CS activity ($P = 0.0008$; Fig 2F).

### 3.2 Changes in the contribution of primary energy supply pathways and steps in honeybees across age and bee castes

In head tissue, NADH-pathway contribution to respiration was lower in queens compared to workers, but significantly only in early-life ($P = 0.030$; Fig 3A). The reduction caused a more pronounced increase in respiration following succinate addition to NADH-pathway in queens compared to the workers, for both early- ($P = 0.040$) and late-life ($P = 0.0007$; Fig 3B).

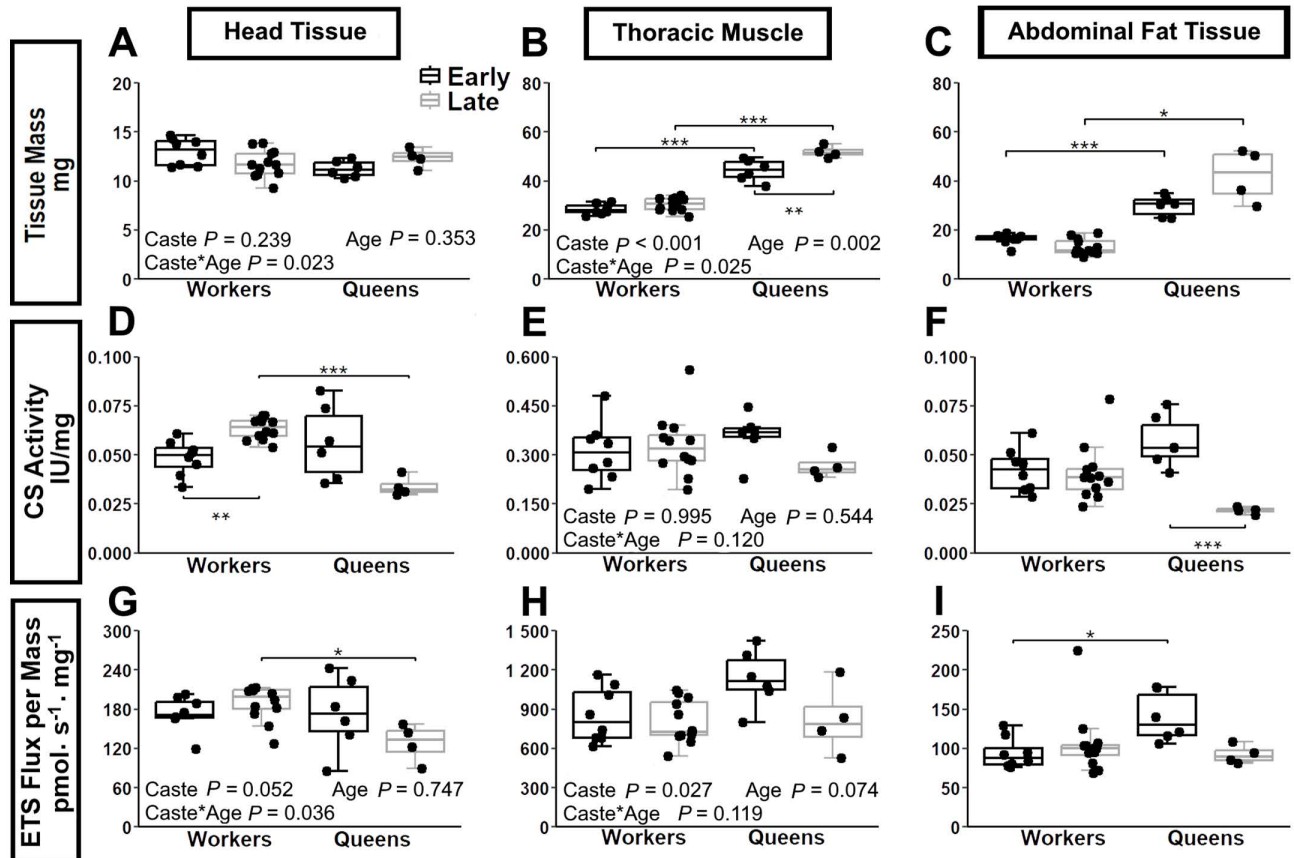

**Fig 2. Tissue mass and biomarker of mitochondrial content in worker and queen bees across different ages and body regions.** Panels A–C report tissue mass (mg), panels D–F quantify citrate synthase (CS) activity as a mitochondrial content marker, and panels G–I present maximal electron transport system (ETS) capacity. All measures are shown for head (A, D, G), thoracic muscle (B, E, H), and abdominal fat body (C, F, I). CS activity and ETS capacity values are normalized to tissue mass (per mg). Box plots display the minimum, 25th percentile, median, 75th percentile, and maximum values. Data represent workers at 1 week (early-life, $n = 8$ and 4 weeks (late-life, $n = 12$ and queen at 1 week (early-life, $n = 6$) and 109 weeks (late-life, $n = 4$). Two-way ANOVA $p$-values for the effects of bee caste (workers vs. queens), age, and their interaction are indicated in each panel. Panel C, D, F and I did not meet the assumptions; therefore, Welch's ANOVA with Games-Howell post-hoc test was performed instead. Significant differences between ages within a bee caste (shown below the boxes) and between bee castes at the same age (shown above the boxes) are denoted * ($P \le 0.05$), ** ($P \le 0.01$), and *** ($P \le 0.001$).

When NS-pathway were combined, there was no significant difference in contribution to maximal flux between queens and workers at any age (Fig 3C). Adding Gp to NS substrates increased respiration across all groups, with a stronger effect in early-life queens (median [min–max]: 0.38 [0.31–0.43]) compared to the three other groups (0.24 [0.06–0.38]), resulting in a significantly higher response in young queens vs. young workers ($P = 0.016$; Fig 3D). This led to greater SGp-pathway contribution in queens compared to workers (early-life: P = 0.007; late-life: $P = 0.005$; Fig 3E). SGp-pathway contribution decreased with age in workers ($P = 0.007$), but not in the queens ($P = 0.484$; Fig 3E). Complex IV activity relative to the maximal ETS flux (Fig 3F) was unchanged with age in both castes (workers: $P = 0.781$; queens: $P = 0.267$), but was significantly higher in late-life queens compared to late-life workers ($P = 0.010$).

In thoracic muscle, no age- or caste-related differences were observed in the NADH-pathway contribution (Fig 4A), the effect of succinate addition to NADH pathway (Fig 4B), the combined NS-pathway (Fig 4C), the effect of Gp addition to NS-pathway (Fig 4D), or the combined SGp-pathway (Fig 4E). Complex IV activity relative to the maximal ETS flux

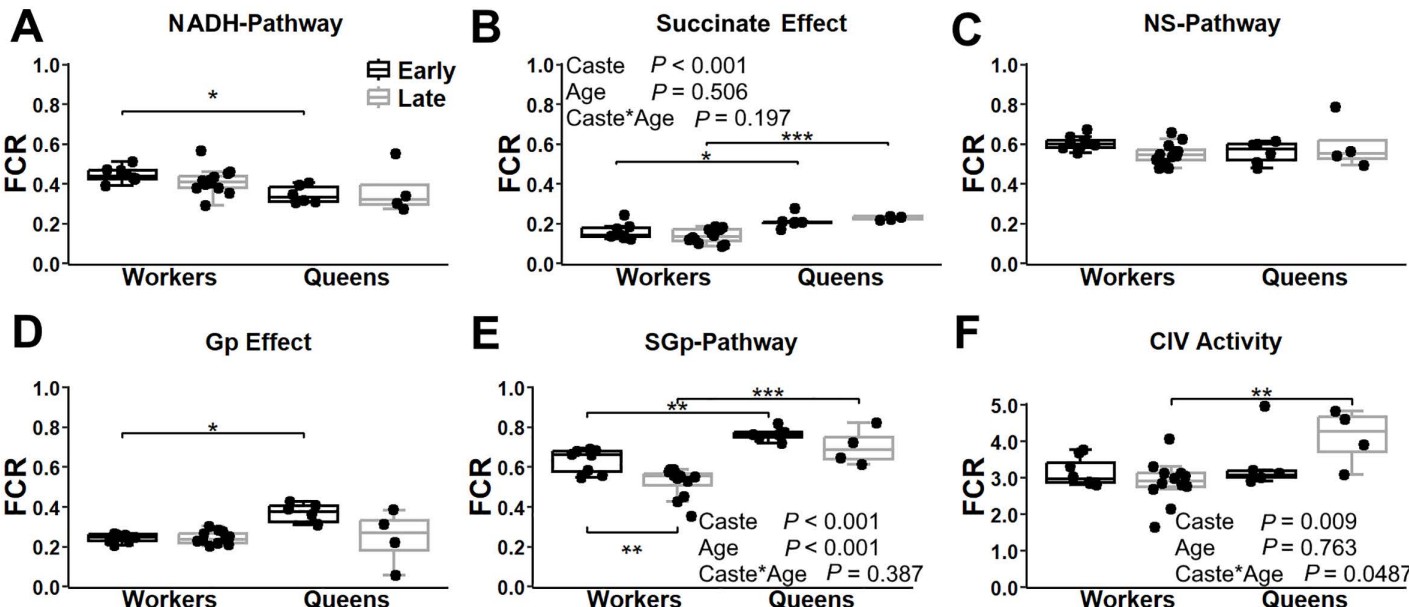

**Fig 3. Mitochondrial respiration pathway contributions in the head tissue of worker and queen bees across ages.** Respiration is expressed as the flux control ratio (FCR), normalized to the maximal electron transport system (ETS) capacity after uncoupling, based on the combined flux of the NADH, Succinate and glycerophosphate (Gp) pathways. Panel A: Relative contribution of the NADH-pathway to maximal flux. Panel B: Increase in respiration following the addition of succinate to mitochondria in the presence of NADH (NS-pathway – NADH-pathway). Panel C: Respiration with combined NS-pathway. Panel D: Effect of Gp addition to mitochondrial respiration in the presence of NS-pathway substrates (NSGp-pathway – NS-pathway). Panel E: Respiration after inhibition of the NADH-pathway with rotenone, with SGp-pathway remaining active. Panel F: Complex IV activity. Box plots display the minimum, 25th percentile, median, 75th percentile, and maximum values. Data represent workers at 1 week (early-life, $n=8$) and 4 weeks (late-life, $n=12$) and queen at 1 week (early-life, $n=6$) and 109 weeks (late-life, $n=4$). Two-way ANOVA $p$-values for the effects of bee caste (workers vs. queens), age, and their interaction are indicated in each panel. Panels A, C and D did not meet the assumptions; therefore, Welch's ANOVA with Games-Howell post-hoc was performed for panel C and Kruskal-Wallis test with Dunn's test was performed for panels A and D instead. Significant differences between ages within a bee caste (shown below the boxes) and between bee castes at the same age (shown above the boxes) are denoted * ($P \leq 0.05$), ** ($P \leq 0.01$), and *** ($P \leq 0.001$).

(Fig 4F) was unchanged with age in queens ($P=0.998$), but showed a decreasing trend in workers ($P=0.064$), resulting in a significantly higher activity in early-life workers compared to early-life queens ($P=0.039$).

In abdominal fat tissue, NADH-pathway contribution remained constant with age in workers ($P=0.531$) but increased from early to late life in queens ($P=0.045$; Fig 5A). This shift was not accompanied by a concomitant change in the effect of succinate addition. In workers, the effect of succinate addition (following NADH-pathway stimulation) declined with age ($P < 0.001$), resulting in similar impacts between older workers and queens at both life stages (Fig 5B). NS-pathway contribution to maximal flux (Fig 5C) decreased with age in workers ($P < 0.001$), and increased in queens ($P < 0.001$). The decline in NS-pathway with age in workers is mostly driven by a reduction in the Succinate-pathway, while the age-related increase in queens was explained by an enhanced NADH-pathway contribution. Adding Gp to NS substrates significantly increased respiration across all groups, with a stronger effect in early-life queens vs. early-life workers ($P=0.012$) or late-life queens ($P=0.046$; Fig 5D). This Gp-induced increase did not vary with age in workers ($P=0.138$). SGp-pathway contribution showed no significant variation with age or caste (Fig 5E). Complex IV activity relative to maximal ETS flux (Fig 5F) was unchanged with age in workers ($P=0.873$), but trended toward an increase with age in queens ($P=0.092$), and was significantly higher in late-life queens vs. late-life workers ($P=0.037$).

Leak-state respiration, expressed as the FCR (a measure of coupling-control and mitochondrial preparation quality) showed no significant differences between caste or age groups in any tissue ($P=0.247$ for head tissue, $P=0.066$ for

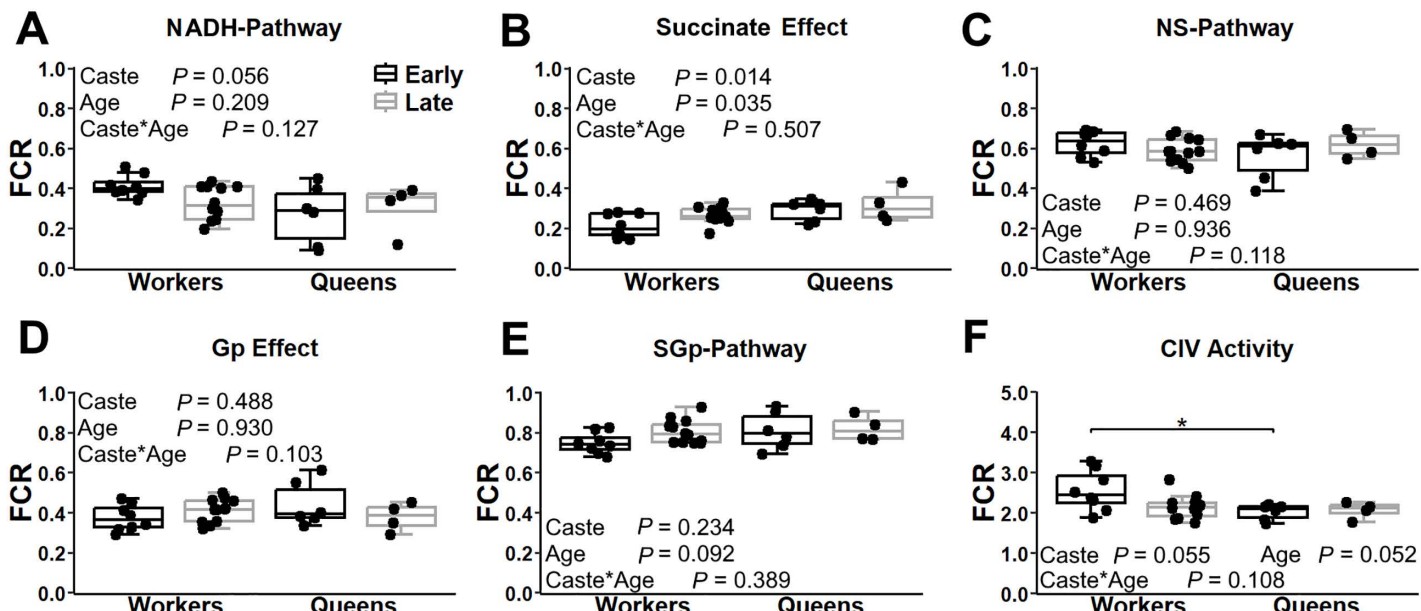

**Fig 4. Mitochondrial respiration pathway contribution in the thoracic muscle of worker and queen bees across ages.** Respiration is expressed as the flux control ratio (FCR), normalized to the maximal electron transport system (ETS) capacity after decoupling, as described in Fig 3. Panels show: (A) NADH-pathway contribution, (B) increase in flux following succinate addition (NS-pathway – NADH-pathway), (C) combined NS-pathway respiration, (D) effect of glycerophosphate addition (NSGp – NS), (E) respiration after addition of rotenone (SGp-pathway), and (F) complex IV activity. Box plots display the minimum, 25th percentile, median, 75th percentile, and maximum values. Sample sizes; workers at 1 week ($n=8$) and 4 weeks ($n=12$); queen at 1 week ($n=6$) and 109 weeks ($n=4$). Two-way ANOVA $p$-values for caste, age, and their interaction are indicated. Significant differences within caste (below boxes) or between bee castes at a given age (above boxes) are denoted * ($P \leq 0.05$), ** ($P \leq 0.01$), and *** ($P \leq 0.001$).

thoracic muscle, $P=0.219$ for abdominal fat tissue), indicating consistent mitochondrial quality. FCR values (median [range]) were very low in head tissue (0.014 [0.000–0.085]), thoracic muscle (0.000 [0.000–0.011]) and abdominal fat tissue (0.035 [0.005–0.080]), reflecting highly efficient mitochondrial coupling.

### 3.3 Changes in the control of OXPHOS by the phosphorylation pathway in honeybees across age and bee castes

Fig 6 shows OXPHOS oxygen consumption via the NSGp-pathway, expressed as a ratio of maximal respiration (ETS capacity) after DNP titration. This ratio reflects the extent to which OXPHOS is limited by the phosphorylation pathway: a value of 1.0 indicates no limitation, while 0.0 represents complete limitation [50,53]. In thoracic muscle, OXPHOS was not limited by the phosphorylation pathway, regardless of caste or age (Fig 6B). In contrast, significant OXPHOS limitations were found in head (Fig 6A) and abdominal fat tissues (Fig 6C). In the head, early-life queens exhibited less OXPHOS limitation than early-life workers ($P=0.033$). Although OXPHOS limitation tended to increase with age in both castes, the trend was not statistically significant (Fig 6A). In abdominal fat, workers showed a clear age-related increase in OXPHOS limitation ($P < 0.001$), whereas queens maintained consistent levels across age groups ($P=0.885$).

### 3.4 Contribution of the fatty acid oxidation pathway to OXPHOS capacity in honeybee tissues

To accurately interpret Gp-pathway contribution to OXPHOS capacity, it is crucial to assess fatty acid oxidation across honeybee tissues. Our results revealed limited to no ability to oxidize palmitoylcarnitine (S2 Fig A) or octanoylcarnitine (S2 Fig B) in the thoracic muscle and head tissue. In contrast, both fatty acid substrates played a significant role in supporting OXPHOS capacity in abdominal fat tissue.

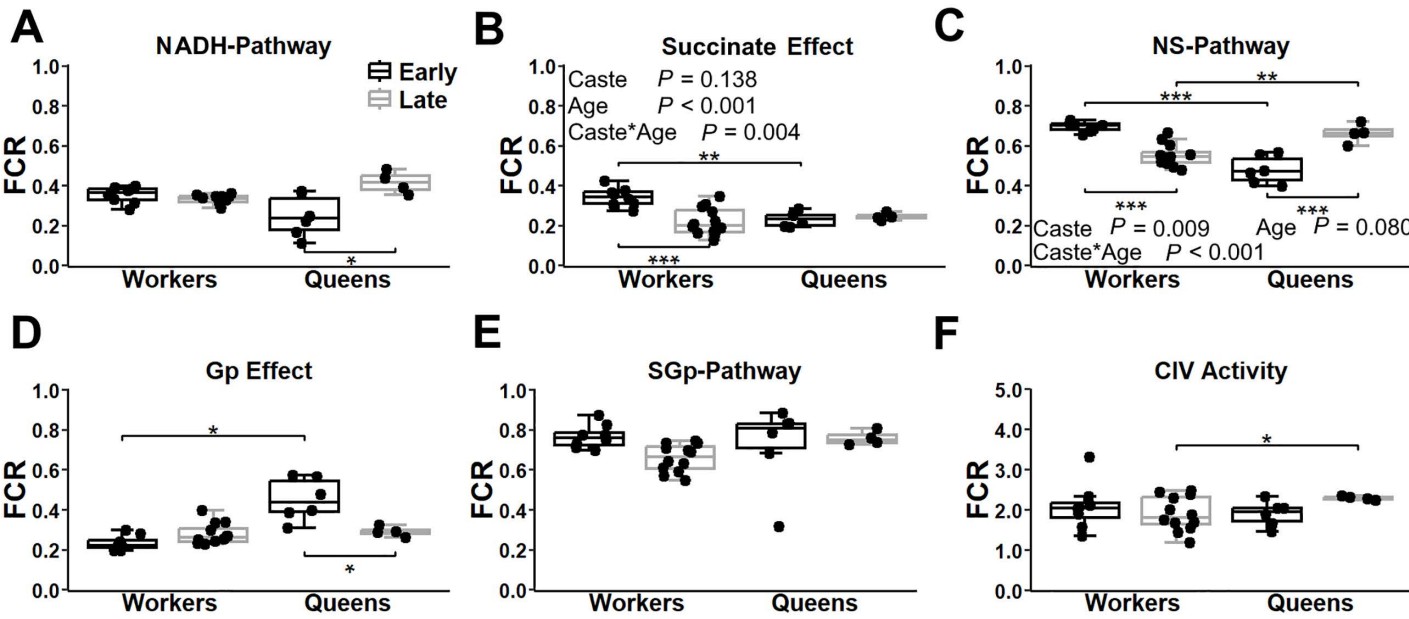

**Fig 5. Mitochondrial respiration pathway contributions in the abdominal fat tissue of worker and queen bees across ages.** Respiration is expressed as the flux control ratio (FCR), normalized to the maximal electron transport system (ETS) capacity after decoupling, as described in Fig 3. Panels show: (A) NADH-pathway contribution, (B) increase in flux following succinate addition (NS-pathway – NADH-pathway), (C) combined NS-pathway respication, (D) effect of glycerophosphate addition (NSGp – NS), (E) respiration after addition of rotenone (SGp-pathway), and (F) complex IV activity. Box plots display the minimum, 25th percentile, median, 75th percentile, and maximum values. Sample sizes; workers at 1 week ($n = 8$) and 4 weeks ($n = 12$); queen at 1 week ($n = 6$) and 109 weeks ($n = 4$). Two-way ANOVA $p$-values for caste, age, and their interaction are indicated. Significant differences within caste (below boxes) or between bee castes at a given age (above boxes) are denoted * ($P \le 0.05$), ** ($P \le 0.01$), and *** ($P \le 0.001$).

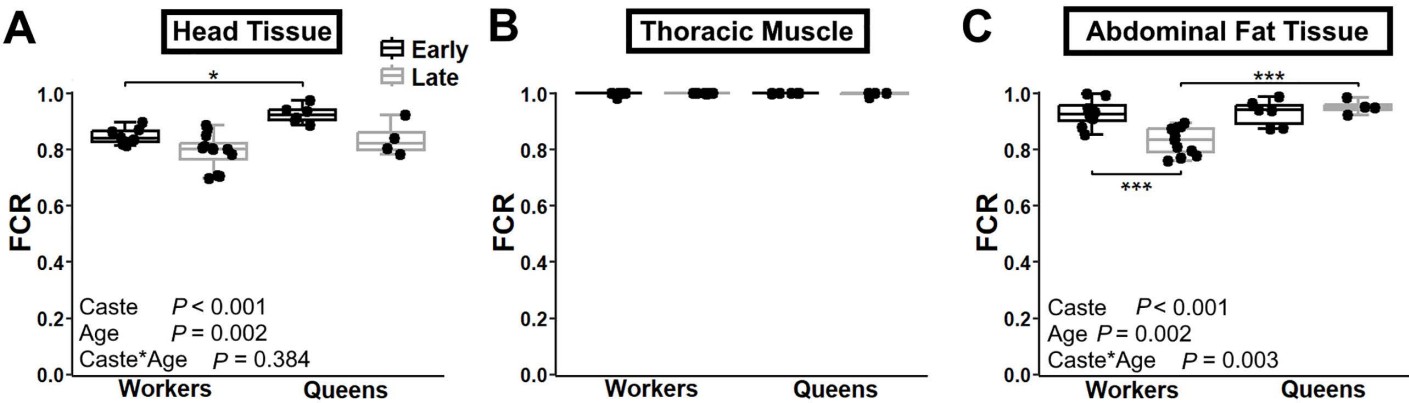

**Fig 6. Control of OXPHOS by the phosphorylation pathway in worker and queen bees across different ages and body regions.** Respiration with combined NSGp-pathways active, prior to decoupling with DNP, illustrating the limitation of OXPHOS by the phosphorylation pathway (1 indicates no limitation and 0 indicates 100% limitation), for the head tissue (A), thoracic muscle (B) and abdominal fat tissue (C). Box plots display the minimum, 25th percentile, median, 75th percentile, and maximum values. Data represent workers at 1 week (early-life, $n = 8$) and 4 weeks (late-life, $n = 12$) and queen at 1 week (early-life, $n = 4$) and 109 weeks (late-life, $n = 4$). Two-way ANOVA $p$-values for the effects of bee caste (workers vs. queens), age, and their interaction are indicated in each panel. Panel B did not meet the assumptions; therefore, Kruskal-Wallis test with Dunn's test was performed instead. Significant differences between ages within a bee caste (shown below the boxes) and between bee castes at the same age (shown above the boxes) are denoted * ($P \le 0.05$), ** ($P \le 0.01$), and *** ($P \le 0.001$).

## 4. Discussion

Although honeybee queens and workers share an identical genome, they exhibit striking lifestyle divergence: queens remain highly fertile and live up to ten times longer than workers. Uncovering the metabolic mechanisms underlying this exceptional longevity and divergent life history remains a key challenge. Our results show that mitochondrial function changes with age in a caste- and tissue-specific manner, highlighting subtle but potentially critical differences that may influence aging trajectories. In the thoracic muscle, despite large differences in energy demands between castes, mitochondrial content and OXPHOS pathway distributions remained consistent across age and caste. This suggests that energy supply in flight muscles is tightly regulated to meet consistent functional demands regardless of lifestyle or lifespan. By contrast, the head and abdominal fat tissue showed caste- and age-related shifts in OXPHOS pathway use: 1) the NADH-pathway contribution was low in young queens, increasing with age in abdominal fat but not in the head, 2) younger queens showed a larger contribution of OXPHOS by the Gp-pathway in head and abdominal fat compared to workers, 3) OXPHOS limitation by the phosphorylation pathway was reduced in queens compared to workers in head tissue (early-life) and abdominal fat tissue (late-life). These nuanced shifts in mitochondrial function suggest bioenergetics plasticity that may help queens maintain cellular energy balance and reduce damage accumulation, contributing to reproductive success and longevity. Overall, our findings suggest that pathway-, age- and tissue-specific differences in mitochondrial function are critical to consider when investigating the biology of aging. Without a deeper understanding of these fine-scale metabolic variations, the connection between mitochondrial function and longevity cannot be fully assessed. This underscores the value of the honeybee as a model system for dissecting how social roles and environmental factors shape metabolic mechanisms underlying lifespan differences.

### 4.1 To Preserve or not to preserve? Mitochondrial content variation across tissues, ages, and castes

Queens had significantly larger thoracic and abdominal tissue masses than workers. Unlike workers, queens exhibited an age-related increase in thoracic mass, with a similar trend in abdominal mass. This tissue growth with age in queens was accompanied by a trend toward reduced mitochondrial content in thoracic muscle and a significant reduction in abdominal fat tissue. These findings align with Darveau *et al.* [54], who reported a negative correlation between body mass and mitochondrial content in male orchid bees, although their study did not address caste-specific differences. Despite queens having substantially larger thorax and abdomen masses, their mitochondrial content per milligram of tissue was similar to that of workers. This suggests a complex metabolic reorganization that compensates for increased body mass and highlights the key role of mitochondrial function in supporting reproductive physiology.

The consistent mitochondrial concentration in queen thoracic muscle may reflect an adaptive strategy to sustain high flight capacity despite increased body mass. Queens face two major flight demands under strong selective pressure: mating flights within 5–10 days post-emergence, and swarming flights later in life to establish a new hive [30]. Both are essential for colony reproduction. The early- and late-life stage examined in our study correspond to these behaviors. The stable mitochondrial content in thoracic muscle, despite increased tissue mass, aligns with prior findings [55]. This suggests that, although queens are less active than workers, they maintain mitochondrial density in their flight muscles, ensuring sufficient bioenergetic capacity during these critical reproductive phases.

The greater abdominal mass in queens compared to workers likely reflects a higher proportion of fat body tissue, which is mainly located in the abdomen. This multifunctional tissue, made up of trophocytes and oenocytes, supports metabolism, nutrient storage, hormone synthesis, immunity, and detoxification [39]. In *A. mellifera* queens, the fat body is especially adapted for reproductive functions [56]. Mitochondrial content in abdominal fat, estimated via CS activity (a well recognized marker of mitochondrial content [45]), remained stable in workers but was initially higher in queens and declined sharply with age. This pattern agrees with Santos *et al.* [57], who found higher mitochondrial gene expression in

queen larvae fat bodies compared to workers. Together, these results suggest that queens undergo early developmental programming for a more metabolically active fat body, which diminishes with age.

Our results showed no significant difference in head size between queens and workers. However, because queens have larger bodies, their brains make up a smaller proportion of total body volume. This aligns previous findings that workers have more developed brain regions [58]. These differences are expected because workers perform cognitively demanding tasks, while queens focus on reproduction and pheromone production, which require less neural processing. The increase in mitochondrial content in worker brains over time—contrasting with a decline in queens—further reflects these distinct roles and supports the absence of functional senescence in aging workers [59].

## 4.2 Dissecting OXPHOS function across tissues, ages, and castes: More than just mitochondrial content

A comprehensive understanding of mitochondrial function extends beyond their abundance in a given tissue; it also involves analyzing the specific relative capacities of OXPHOS pathways. Such analysis reveals several meaningful insights into how mitochondrial energy production is modulated across tissues, castes and ages in honeybees. Our first key finding emerged from the thoracic muscle, where the relative contributions of the conventional NADH- and succinate-linked pathways, as well as the Gp-pathway—critical for insect flight muscle function (reviewed by Mráček, Drahota and Houštěk [44])— remained remarkably consistent across age and castes. These results may appear to contrast with findings from Menail et al. [36] and Cervoni et al. [34], who reported age-related increases in mass-specific respiration in honeybee thoracic muscle. Two main factors explain this discrepancy. First, Menail et al. [36] normalized their data to tissue mass. However, when FCRs were recalculated using their deposited raw data, a markedly different age-dependent pattern emerged. Instead of a gradual increase with age across all pathways, FCRs remained relatively stable across age groups. In line with our findings, no statistically significant differences were observed between 8-day-old individuals and 25-day-old foragers for any pathway [e.g., NADH pathway 0.26 (0.18–0.34) at day 8 vs. 0.37 (0.22–0.56) at day 25]. *FCR* from a single respirometric run with sequential titrations provide an internal normalization, reflecting respiratory control independently of mitochondrial content [51]. Thus, the flux per mass differences observed in Menail et al. [36] are largely explained by age-related changes in mitochondrial content, as indicated by both their data of CS activity data and ours. Furthermore, Menail et al. [36] reported a significant increase in coupling efficiency with age. This highlights the need for caution when interpreting age-related changes, as mitochondrial integrity in younger bees may have been compromised during sample preparation. In contrast, our study found consistently lower and stable FCR for leak across groups, indicating well preserved mitochondrial function and coupling regardless of age or caste. In Cervoni et al. [34], the minimal increase in respiration following ADP addition likely reflects restricted mitochondrial access as the tissue was neither homogenized nor permeabilized. Because solid tissues contain multiple cell layers and extracellular matrices that limit diffusion of oxygen, ADP, and metabolic substrates, mitochondrial respiration is typically assessed in isolated mitochondria, permeabilized preparations, or homogenates [60] where mitochondria have unrestricted access to substrates, ADP, and oxygen. These results underscore the importance of normalization and tissue preparation when interpreting mitochondrial data. Overall, our findings emphasized that the consistent proportional contributions of mitochondrial pathways in thoracic muscle were not incidental but reflect a biologically meaningful pattern of tightly conserved energy metabolism essential for maintaining flight capability across castes and ages.

Another key insight from our analysis of OXPHOS pathway capacities was the caste specific organization of the NADH-pathway in head and abdominal fat tissue, unlike in thoracic muscle. In head tissue, early-life queens relied less on the NADH-pathway than early-life workers. In abdominal fat tissue, queens showed an age-related increase in NADH-pathway contribution, whereas workers showed no such change. This pattern – reduced early life NADH-pathway capacity in long-lived queens – aligns with Mast et al. [13], who showed that seed beetles selected for delayed reproduction and extended lifespan also had lower early-life NADH-pathway contribution. Indeed, across species and taxa, reduced complex I capacity, content or gene expression due to natural variation or experimental inhibition – particularly

early in life – has been repeatedly linked to extended longevity [7,11,14,17,18,20–22,61,62]. Altogether, our results suggested that queen physiology reflects a strategic balance: maintaining low NADH-pathway capacity early in life to promote longevity, while meeting essential metabolic demands through alternative pathways. The NADH-pathway involves three proton pumping complexes, compared to only two in the Succinate- or Gp-pathway, enabling higher ATP yield. Consequently, reduced reliance on the NADH-pathway and increased Succinate-pathway contribution in queen head tissue may be harmless, given their reduced brain activity compared to workers heavily engaged in memory, navigation, communication, and social interactions [58]. In the abdominal fat, queens, unlike workers, prioritized a low early-life NADH-pathway contribution despite the high energy demand of reproduction, such as oogenesis and pheromone biosynthesis. This consistent early-life downregulation of the NADH-pathway in queen head and abdominal fat tissue may represent a key strategy supporting their exceptional longevity. These pathway differences are likely influenced by caste-specific adult diets. Workers rely predominantly on carbohydrate-rich foods (nectar and honey) to fuel their high activity levels, whereas queens are continuously fed royal jelly, a secretion rich in proteins and lipids provided by nurse workers [30]. This nutrient supply favors the use of non-carbohydrate substrates in queens and may promote greater engagement of alternative mitochondrial pathways. Such metabolic flexibility could support the queens' sustained reproduction and contribute to their remarkable lifespan.

The Gp-pathway represents a central component of this metabolic flexibility in queens. It underwent significant age- and caste-related changes in both head and abdominal fat tissue. This pathway is central to the glycerophosphate shuttle, linking glycolysis, OXPHOS, gluconeogenesis, and lipid anabolism and catabolism [44]. Gp is derived from two primary sources: [1] glycerol from triacylglycerols breakdown, and [2] dihydroxyacetone phosphate (DHAP), a glycolytic intermediate reduced to Gp by cytosolic Gp dehydrogenase (cGpDH), regenerating $NAD^+$ to sustain high glycolytic flux [44]. Once formed, Gp enters mitochondria, where mitochondrial Gp dehydrogenase (mGpDH; located in the inner mitochondrial membrane) transfers electrons from Gp to FAD, then to the ETS at the Q-junction. When needed, Gp can also be redirected towards triglycerides synthesis, providing the glycerol backbone and acting as a rate-limiting factor in lipid biosynthesis. This highlights its essential role in abdominal fat bodies. Therefore, the Gp-pathway is essential not only in tissues with high glycolytic activity but also in those involved in lipid synthesis and catabolism, coordinating energy production and storage. Notably, mGpDH expression is highest in brown adipose tissue, skeletal muscle (including insect flight muscles), and brain (reviewed by Mráček, Drahota and Houštěk [44]).

Two key findings emerged from our analysis of the Gp-pathway's contribution to maximal OXPHOS capacity in honeybees. First, the Gp-pathway played a significant role in the thoracic muscle, contributing approximatively 40% to maximal OXPHOS capacity. This contribution remained consistent across ages and castes, highlighting its importance for muscle function. In these highly active muscles, which showed no evidence of FAO [our results, [38,41,43]], most Gp likely originated from DHAP, a glycolytic intermediate. In this context, the Gp shuttle's primary function is to regenerate $NAD^+$ in the cytosol, supporting high glycolytic rates [40,44]. Previous studies on insects have shown that intense muscle activity, such as flight, increases Gp production while minimizing lactate accumulation [63]. Our findings aligned with this, suggesting that the Gp-pathway in honeybee thoracic muscle helps mGpDH reoxidize cytosolic NADH during glycolysis, thereby preventing lactate build-up in these energy-demanding cells. Second, we found high Gp-pathway's contribution in abdominal fat bodies of young queens. Unlike thoracic muscle, this tissue showed sustained lipid metabolism, including both lipogenesis [64] and FAO [our results, [64]]. Here, Gp likely derived not only from glycolytic intermediates but also from triglyceride breakdown, reflecting the dual role of fat bodies in energy storage and mobilization. Elevated Gp shuttle activity in young queens likely promotes tight metabolic coupling between glycolysis, lipogenesis, and OXPHOS fueled by both carbohydrates and fatty acids. This increased Gp-pathway contribution likely facilitates ATP production under varying metabolic demands by compensating for the reduced NADH-pathway capacity seen in this group. Our results aligned with Mráček, Drahota and Houštěk [44], whom proposed that the GpDH shuttle bypass complex I by oxidizing cytosolic NADH. A similar mechanism may explain the increased Gp pathway contribution in queen head tissue, were NADH-pathway

capacity is also low. Elevated Gp-pathway capacity may be critical for meeting the energetic reproductive demand of queens.

One potential consequence of increased Gp-pathway contribution is elevated production of ROS, primarily due to electron leakage at the Q junction [65]. This ROS-generating potential likely explains why the pathway is suppressed in most tissues [44]. However, a transient increase in ROS early in life may have an adaptive value. Early-life ROS bursts have been linked to increased longevity in *C. elegans* [66] and *D. melanogaster* [14], possibly via activation of mitochondrial unfolded protein response [14,22]. Therefore, early-life upregulation of the Gp-pathway might promote long-term resilience and healthy aging. So even considering the disadvantage of the Gp-pathway being partially uncoupled from ATP synthesis [42], it offers important metabolic flexibility by enabling (i) reoxidation of cytosolic NADH in glycolytic cells; (ii) bypass of mitochondrial complex I; and (iii) regulation of cytosolic Gp, a key metabolite linking glycolysis, lipogenesis, and OXPHOS from both glucose and lipid and sources [44].

### 4.3 Phosphorylation pathway constraints on OXPHOS: An often overlooked but potentially significant factor

To fully assess mitochondrial output, it was essential to evaluate potential OXPHOS limitations imposed by the phosphorylation pathway, which includes the ATP synthase, the adenine nucleotide translocator, and the phosphate carrier. Together, these components convert the protonmotive force into usable ATP. Although rare in animals, such limitation has been measured in human heart [50] and skeletal muscle [52], horse skeletal muscle [67], and planarian [53]. Despite being frequently overlooked, this limitation is a critical factor when assessing mitochondrial function as impaired ATP synthesis, despite intact electron transport, can lead to increased electron leakage and ROS production [68]. Our results revealed significant variation in the OXPHOS limitation by the phosphorylation pathway across tissues, castes and ages. The first notable finding was the absence of limitation in thoracic muscle, consistent with the high energy demands and preserved mitochondrial function in this tissue across age and caste. In abdominal fat, bees also maintained very low limitation, though it increased with age in workers while remaining low in queen. This could potentially help limit ROS production in aging queens, and reflects the sustained energy demands of reproduction. In head tissue, OXPHOS limitation by the phosphorylation pathway tended to decrease with age in both castes. However, queens started life with a significantly lower limitation compared to workers, so by the end of life, values in both castes were equivalent. This may represent an adaptive mechanism limiting ROS production throughout queens' lives. Whether this is part of the aging process warrants further investigation.

Our study reveals clear caste- and age-related differences in mitochondrial function. The significant interaction between caste and age indicates that age-related changes in mitochondrial function differ between queens and workers. However, we have not directly linked these differences to the castes' distinct life expectancies or aging rates. Caste differences in aging have previously been attributed to a complex interplay of intrinsic and extrinsic factors [69], and it is important to acknowledge that queens and workers differ in multiple aspects, including ontogeny, behavior, reproductive status, lifestyle, and environmental exposures. This complexity makes it difficult to attribute caste-specific aging solely to the mitochondrial differences we describe [70]. Nonetheless, our findings suggest a plausible mechanistic contribution to caste-specific aging, especially given the well-established role of mitochondria in aging processes [71]. By identifying tissue- and pathway-specific shifts in mitochondrial function within a single genome, our results provide novel insight into how metabolism may modulate lifespan. This work lays the foundation for future experiments aimed at disentangling the relative contributions of developmental programming, lifestyle, and environmental factors.

### 4.4 Concluding remarks

Our study presents a comprehensive analysis of how specific OXPHOS pathways vary across life stages and tissues of short-lived worker and long-lived queen honey bees, integrating cross-sectional and longitudinal comparisons. The mitochondrial phenotypes observed reflect a high degree of physiological plasticity, likely shaped by distinct life-history

strategies and energetic demands associated with each caste. Altogether, our results underscore the profound metabolic plasticity in eusocial insects, where mitochondrial remodeling appears to support caste-specific adaptations by balancing performance and cellular maintenance. While we cannot directly establish causal connections between our measures and different rates of ageing, the observed patterns are consistent with the idea that mitochondrial plasticity influences physiological resilience and modulates ageing in organisms derived from a shared genome. These findings are the initial steps towards a broader understanding of how specific metabolic processes can evolve in response to social structure and differential selection for longevity that depends on specialized social roles.

### Highlights

- Tissue- and caste-specific mitochondrial traits link to queen longevity.

- Distinct mitochondrial patterns between castes emerge early in life.

- Mitochondrial shifts localize to head and abdominal fat, not thoracic muscle.

- Queens pivot metabolism: boosting glycerophosphate flux to offset low NADH pathway.

### Supporting information

**S1 Fig. Complex IV activity as an additional biomarker of mitochondrial content in worker and queen bees across different ages and body regions.**
(DOCX)

**S2 Fig. Leak and OXPHOS capacity of fatty acid oxidation pathways in various tissues of worker bees.**
(DOCX)

**S1 File. Graphical abstract.**
(TIF)

### Acknowledgments

We thank Loïc Drda for his assistance with the design of the graphical abstract in S1 File. Language editing assistance was provided using an AI-based tool (ChatGPT) to improve grammar and clarity.

### Author contributions

**Conceptualization:** Clément Joël Lucien Chevret, José Francisco Echegaray, Alexander Walton, Maryam Lo, Olav Rueppell, Hélène Lemieux.

**Data curation:** Clément Joël Lucien Chevret, José Francisco Echegaray, Alexander Walton, Maryam Lo, Olav Rueppell, Hélène Lemieux.

**Formal analysis:** Clément Joël Lucien Chevret, José Francisco Echegaray, Alexander Walton, Maryam Lo, Hélène Lemieux.

**Funding acquisition:** Olav Rueppell, Hélène Lemieux.

**Methodology:** Clément Joël Lucien Chevret, José Francisco Echegaray, Olav Rueppell.

**Resources:** Olav Rueppell.

**Supervision:** Olav Rueppell, Hélène Lemieux.

**Validation:** José Francisco Echegaray, Alexander Walton, Olav Rueppell, Hélène Lemieux.

**Visualization:** Maryam Lo.

**Writing – original draft:** Clément Joël Lucien Chevret, Hélène Lemieux.

**Writing – review & editing:** Clément Joël Lucien Chevret, José Francisco Echegaray, Alexander Walton, Maryam Lo, Olav Rueppell.

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
