## [Decision Letter · Decision Letter 0]

29 Oct 2025

Dear Dr. Lemieux,

Thank you for submitting your manuscript to PLOS ONE. After careful consideration, we feel that it has merit but does not fully meet PLOS ONE’s publication criteria as it currently stands. Therefore, we invite you to submit a revised version of the manuscript that addresses the points raised during the review process.

We look forward to receiving your revised manuscript.

Kind regards,

Wolfgang Blenau

Academic Editor

PLOS ONE

**Journal Requirements:**

“This study was supported by the following grants: two Discovery Grants from the Natural Sciences and Engineering Research Council of Canada (RGPIN-2021-02924 to H. Lemieux and RGPIN-2022-03629 to O. Rueppell), and a grant from the U.S. Army Research Office (W911NF-22-1-0195 to O. Rueppell).”

4. Please note that your Data Availability Statement is currently missing the repository name and or a direct link to access each database. If your manuscript is accepted for publication, you will be asked to provide these details on a very short timeline. We therefore suggest that you provide this information now, though we will not hold up the peer review process if you are unable.

6. We are unable to open your Supporting Information file “fig_s1 and fig_s2. Please kindly revise as necessary and re-upload.

**Additional Editor Comments:**

As you will see below, the two reviewers' opinions differ greatly. In order not to waste any more time obtaining a third opinion, I have tried to find a compromise with the recommendation “Major Revision.” I would ask you to take the methodological concerns of Reviewer #2 very seriously.

Both reviewers have problems with the normalization step as performed in this study. Reviewer #2 also notes that critical tissue types should have been prepared more carefully. Furthermore, more specific information on the selection and status of the animals used is required. In addition, reviewer #1 makes a number of constructive suggestions for improving the introduction and requests additional details in the description of the methods. Reviewer #1 makes some suggestions for deepening the discussion.

Reviewers' comments:

Reviewer's Responses to Questions

**Comments to the Author**

1. Is the manuscript technically sound, and do the data support the conclusions?

Reviewer #1: Yes

Reviewer #2: No

2. Has the statistical analysis been performed appropriately and rigorously?

Reviewer #1: Yes

Reviewer #2: I Don't Know

3. Have the authors made all data underlying the findings in their manuscript fully available?

Reviewer #1: Yes

Reviewer #2: Yes

4. Is the manuscript presented in an intelligible fashion and written in standard English?

Reviewer #1: Yes

Reviewer #2: No

Reviewer #1: This study presents high-resolution respirometry results comparing young vs. old queen and worker honey bees, with the implicit aim of gaining insights into the possible role of mitochondrial function (OXPHOS pathways) in shaping the immense lifespan difference between the castes of social insects. After comparing body compartment size and of citrate synthase activity as a proxy of functional mitochondrial units, the experiments were performed on homogenates of head (mainly brain), thorax (mainly flight muscle) and abdominal (mainly fat body) tissues using an Oroboros system to measure the age-, caste-, and tissue-specific rates of mitochondrial respiration. The experiments were carefully performed and the protocols are described in sufficient detail to serve as guidance for follow-up studies building on this approach and the obtained results. The detailed methods description is also of importance readers not familiar with this methodology, so that they can capture and understand the results and the respective conclusions.

This said, I only have a view comments and questions to this otherwise well-prepared manuscript.

1) A critical issue in presenting comparative HRR data is the normalization step. As far as I understand, the flux control ratios (FCRs) were calculated by normalization against the non-mitochondrial residual oxygen consumption (ROX). This, however, does not take into account the changes in functional mitochondrial units along the bees’ lifespan and tissue type, as shown in Figure 1, where this data is reported as citrate synthase activity, and where differences are clearly apparent for the head and abdomen. I certainly would expect that this factor can affect the interpretation of the results.

2) The authors compared their results to those of two prior studies that used a similar approach, and they explain why their results differ from those of these prior studies. While I agree with the argument for the first study (Menai et al.), the second study (Cervoni et al.) was criticized for having been performed in intact tissue and not in a homogenate. This, however, relates to the question whether a study examining the cellular physiology should be performed on intact cells in a tissue context, or whether a homogenate with mitochondria separated from the cellular context would be better representation. Both have drawbacks, on the one hand less control over the accessibility of the added compounds to the (unpermeabilized) mitochondria, while on the other, the cytoplasmic context is lost.

3) One issue that is of relevance in bioenergetics and metabolism is the type of diet consumed by the experimental animals. In the case of honey bees, there is a major difference in the diet consumed by young workers (high protein/lipid) compared to old workers (essentially only carbohydrate) and to queens, where both young and old queens receive royal jelly (high protein/lipid) from the workers. Adding a sentence or two on this would enrich the discussion.

4) The legends for Figures 2-4 are very similar, showing the datasets separately for head, thorax, and abdominal tissues. Hence, for Figures 3 and 4 the legend text could be condensed.

5) Unfortunately I could not access the data deposited in DRYAD, but I trust that they will be made available upon publication.

Reviewer #2: The goal of PLoS ONE is to publish high quality studies irrespective of novelty in the field. I found that this study addresses an important and interesting (and novel) topic, but the quality of the study and presentation have significant limitations. I have substantial concerns about the methodology; as it was carried out, the approach limits the interpretability of the data. Some of these limitations are clear, and there are other places where the Methods are not complete enough to interpret the results. I focused my comments on the Introduction and Methods, because in my opinion, these substantial concerns would need to be addressed before it is worth addressing the Results and Discussion. I provide details about my concerns in the attached document.

**Do you want your identity to be public for this peer review?** For information about this choice, including consent withdrawal, please see our Privacy Policy

Reviewer #1: No

Reviewer #2: No

---

## [Author Response · Author response to Decision Letter 1]

8 Dec 2025

The response has been included in a more readable format within the cover letter.

Journal requirements

We have carefully reviewed the PLOS ONE formatting and file-naming guidelines and have revised the manuscript and all associated files to ensure full compliance with the journal’s style requirements. The figures are now named properly on the files and in the manuscript.

2. Thank you for stating the following financial disclosure: “This study was supported by the following grants: two Discovery Grants from the Natural Sciences and Engineering Research Council of Canada (RGPIN-2021-02924 to H. Lemieux and RGPIN-2022-03629 to O. Rueppell), and a grant from the U.S. Army Research Office (W911NF-22-1-0195 to O. Rueppell).”

We revised the Funding section to include the following statement (Lines 632-633): “The funders had no role in study design, data collection and analysis, decision to publish, or preparation of the manuscript.”

There is no additional funding to declare beyond what is listed in the Funding Statement.

4. Please note that your Data Availability Statement is currently missing the repository name and or a direct link to access each database. If your manuscript is accepted for publication, you will be asked to provide these details on a very short timeline. We therefore suggest that you provide this information now, though we will not hold up the peer review process if you are unable.

The data have been deposited in Dryad and are currently under the status “Private for Peer Review.” The dataset will be made publicly accessible upon acceptance of the manuscript.

The following statement have been added (Lines 678-680): ‘’The data underlying this study have been deposited in Dryad and are accessible at https://datadryad.org/share/LINK_NOT_FOR_PUBLICATION/XMuWZub4VmU6WNCn-n8y9EJWIYRmGh9onxOSkTQ24MM.’’

6. We are unable to open your Supporting Information file “fig_s1 and fig_s2. Please kindly revise as necessary and re-upload.

We have uploaded the supporting figures and have made sure they can be accessible now.

Not applicable. No additional citations were recommended by the reviewers.

Additional Editor Comments

As you will see below, the two reviewers' opinions differ greatly. In order not to waste any more time obtaining a third opinion, I have tried to find a compromise with the recommendation “Major Revision.” I would ask you to take the methodological concerns of Reviewer #2 very seriously.

Both reviewers have problems with the normalization step as performed in this study. Reviewer #2 also notes that critical tissue types should have been prepared more carefully. Furthermore, more specific information on the selection and status of the animals used is required. In addition, reviewer #1 makes a number of constructive suggestions for improving the introduction and requests additional details in the description of the methods. Reviewer #1 makes some suggestions for deepening the discussion.

Thank you for the comments. As detailed below, we have carefully reviewed and addressed all points raised by both reviewers, and where a suggestion was not implemented, we have provided a clear rationale.

Reviewer 1

This study presents high-resolution respirometry results comparing young vs. old queen and worker honey bees, with the implicit aim of gaining insights into the possible role of mitochondrial function (OXPHOS pathways) in shaping the immense lifespan difference between the castes of social insects. After comparing body compartment size and of citrate synthase activity as a proxy of functional mitochondrial units, the experiments were performed on homogenates of head (mainly brain), thorax (mainly flight muscle) and abdominal (mainly fat body) tissues using an Oroboros system to measure the age-, caste-, and tissue-specific rates of mitochondrial respiration. The experiments were carefully performed and the protocols are described in sufficient detail to serve as guidance for follow-up studies building on this approach and the obtained results. The detailed methods description is also of importance readers not familiar with this methodology, so that they can capture and understand the results and the respective conclusions.

This said, I only have a view comments and questions to this otherwise well-prepared manuscript.

Thank you for your positive and constructive comments. We have carefully considered each point and addressed them in detail below.

1) A critical issue in presenting comparative HRR data is the normalization step. As far as I understand, the flux control ratios (FCRs) were calculated by normalization against the non-mitochondrial residual oxygen consumption (ROX). This, however, does not take into account the changes in functional mitochondrial units along the bees’ lifespan and tissue type, as shown in Fig 1, where this data is reported as citrate synthase activity, and where differences are clearly apparent for the head and abdomen. I certainly would expect that this factor can affect the interpretation of the results.

The flux control ratios (FCRs) provide a quantitative “fingerprint” of mitochondrial coupling and respiratory control by expressing each respiratory state relative to a common internal reference. In sequential titration protocols, FCRs enable internal normalization of oxygen flux, eliminating the need to correct for mitochondrial content (although we also quantified it here using citrate synthase activity). Each FCR represents the ratio of oxygen flux in a given mitochondrial respiratory state (Yi) to that in a defined reference state (Z), typically the maximal electron transfer system (ETS) capacity. In this study, the reference state corresponds to the uncoupled ETS condition, where NADH-, succinate-, glycerophosphate-, and fatty acid oxidation–linked pathways are all active. In the earlier version of the manuscript, FCRs were defined in the Methods section and figure legends (Figs 2–4). To enhance clarity, we have now added the explicit formula and expanded the explanation to illustrate more clearly how this ratio is calculated and interpreted (Lines 235-240).

‘’Alternatively, respiration was expressed as the flux control ratio [FCR; (39, 44)], defined as the respiration of a specific pathway or complex (Yi) normalized to the rate Z, i.e., the maximal ETS capacity (NSGp- or FAONSGp-pathway).

FCRi = Yi

Z

FCRs, by comparing oxygen flux in different respiratory states to a common reference state, allow comparisons that are independent of mitochondrial content or assay conditions.’’

Furthermore, to better illustrate how each respiratory step contributes to the calculation of its corresponding FCR, we have added a representative trace of the Substrate–Uncoupler–Inhibitor Titration protocol in the Methods section (Fig 1; Legends on Lines 212-222), allowing readers to visualize the sequential titrations, the identification of each respiratory state (Yi), and the point at which the maximal ETS reference state (Z) is reached.

2) The authors compared their results to those of two prior studies that used a similar approach, and they explain why their results differ from those of these prior studies. While I agree with the argument for the first study (Menai et al.), the second study (Cervoni et al.) was criticized for having been performed in intact tissue and not in a homogenate. This, however, relates to the question whether a study examining the cellular physiology should be performed on intact cells in a tissue context, or whether a homogenate with mitochondria separated from the cellular context would be better representation. Both have drawbacks, on the one hand less control over the accessibility of the added compounds to the (unpermeabilized) mitochondria, while on the other, the cytoplasmic context is lost.

A solid tissue has multiple cell layers embedded within extracellular matrices that create a significant diffusion barrier for (1) oxygen1-3, (2) ADP and (3) respiratory substrates.4 Even when these molecules reach the plasma membrane, they cannot freely cross it without specific transporters or membrane permeabilization, since ADP and most mitochondrial substrates (e.g., malate, succinate) are highly polar or charged and therefore impermeant to intact cell membranes without tightly regulated transport mechanisms. Consequently, diffusion into deeper cell layers is severely restricted. When an intact tissue chunk is placed in a respirometry chamber, only the outermost cell layers receive adequate oxygen and substrate supply. The tissue core rapidly becomes hypoxic or even anoxic, and most mitochondria within the inner cells never encounter the added substrates or ADP required to sustain oxidative phosphorylation. As a result, the oxygen consumption recorded reflects mainly a non-uniform, diffusion-limited respiration of the outer tissue layers, rather than the intrinsic mitochondrial capacity of the entire tissue.

For these reasons, mitochondrial respiration assays are routinely performed on isolated mitochondria, intact detached cells (or a single layer of attached cells), permeabilized cells or tissue fibers, or homogenates,5 all of which ensure that mitochondria have unrestricted access to substrates, ADP, and oxygen, enabling accurate assessment of their functional capacity.

The manuscript was revised as follows to provide a clearer explanation:

‘In Cervoni, Cardoso-Júnior (34), the minimal increase in respiration following ADP addition likely reflects limited restricted mitochondrial access due to tissue neither homogenized nor permeabilized. Because solid tissues contain multiple cell layers and extracellular matrices that limit diffusion of oxygen, ADP, and metabolic substrates, mitochondrial respiration is typically assessed in isolated mitochondria, permeabilized preparations, or homogenates (60) where mitochondria have unrestricted access to substrates, ADP, and oxygen.‘’ (Lines 535-540)

References cited above:

1. Pias SC. How does oxygen diffuse from capillaries to tissue mitochondria? Barriers and pathways. The Journal of physiology. 2021;599:1769-1782.

2. Scandurra FM and Gnaiger E. Cell respiration under hypoxia: facts and artefacts in mitochondrial oxygen kinetics. Advances in experimental medicine and biology. 2010;662:7-25.

3. Place TL, Domann FE and Case AJ. Limitations of oxygen delivery to cells in culture: An underappreciated problem in basic and translational research. Free radical biology & medicine. 2017;113:311-322.

4. Pajor AM. Sodium-coupled dicarboxylate and citrate transporters from the SLC13 family. Pflugers Archiv : European journal of physiology. 2014;466:119-30.

5. Doerrier C, Garcia-Souza LF, Krumschnabel G, Wohlfarter Y, Mészáros AT and Gnaiger E. High-Resolution FluoRespirometry and OXPHOS protocols for human cells, permeabilized fibers from small biopsies of muscle, and isolated mitochondria. Methods Mol Biol. 2018;1782:31-70.

3) One issue that is of relevance in bioenergetics and metabolism is the type of diet consumed by the experimental animals. In the case of honey bees, there is a major difference in the diet consumed by young workers (high protein/lipid) compared to old workers (essentially only carbohydrate) and to queens, where both young and old queens receive royal jelly (high protein/lipid) from the workers. Adding a sentence or two on this would enrich the discussion.

We thank the reviewer for this valuable suggestion. We agree that caste- and age-specific diets are an important aspect of honeybee physiology and directly relevant to bioenergetic differences. In response, we have expanded the Discussion to include a dedicated section on the impact of dietary composition (Lines 563–569). The new text reads:

“These pathway differences are likely influenced by caste-specific adult diets. Workers rely predominantly on carbohydrate-rich foods (nectar and honey) to fuel their high activity levels, whereas queens are continuously fed royal jelly, a secretion rich in proteins and lipids provided by nurse workers (30). This nutrient supply favors the use of non-carbohydrate substrates in queens and may promote greater engagement of alternative mitochondrial pathways. Such metabolic flexibility could support queens’ sustained reproductive output and contribute to their remarkable lifespan.”

We believe that this addition strengthens the discussion by linking mitochondrial pathway use with well-established nutritional differences among castes and worker age groups.

4) The legends for Figures 2-4 are very similar, showing the datasets separately for head, thorax, and abdominal tissues. Hence, for Figures 3 and 4 the legend text could be condensed.

We appreciate the reviewer’s suggestion to streamline the figure legends. In response, we have revised the legends for Figs 3 (Lines 333-348), 4 (Lines 357-372) and 5 (Lines 374-384) to reduce redundancy while ensuring that each figure remains fully interpretable on its own. Methodological details and pathway definitions that were repeated across legends have now been condensed, with cross-references to Fig 3 where appropriate. This approach minimizes repetition while preserving clarity and scientific completeness.

5) Unfortunately I could not access the data deposited in DRYAD, but I trust that they will be made available upon publication.

Thank you for noting this. Yes, the data will be made publicly available upon acceptance of the manuscript, in accordance with the journal’s data-sharing policy. We have already prepared the dataset and uploaded it to DRYAD, and it will be fully accessible once the paper is formally accepted.

Reviewer 2

T

---

## [Decision Letter · Decision Letter 1]

5 Jan 2026

Tissue-Specific Mitochondrial Pathway Remodeling Linked to Longevity in Honeybee Queens

PONE-D-25-49904R1

Dear Dr. Lemieux,

We’re pleased to inform you that your manuscript has been judged scientifically suitable for publication and will be formally accepted for publication once it meets all outstanding technical requirements.

Kind regards,

Wolfgang Blenau

Academic Editor

PLOS One

Additional Editor Comments (optional):

Reviewers' comments:

Reviewer's Responses to Questions

**Comments to the Author**

Reviewer #1: All comments have been addressed

2. Is the manuscript technically sound, and do the data support the conclusions?

Reviewer #1: (No Response)

3. Has the statistical analysis been performed appropriately and rigorously?

Reviewer #1: (No Response)

4. Have the authors made all data underlying the findings in their manuscript fully available?

Reviewer #1: (No Response)

5. Is the manuscript presented in an intelligible fashion and written in standard English?

Reviewer #1: (No Response)

Reviewer #1: (No Response)

**Do you want your identity to be public for this peer review?** For information about this choice, including consent withdrawal, please see our Privacy Policy

Reviewer #1: No

---

## [Editor Report · Acceptance letter]

PONE-D-25-49904R1

PLOS One

Dear Dr. Lemieux,

I'm pleased to inform you that your manuscript has been deemed suitable for publication in PLOS One. Congratulations! Your manuscript is now being handed over to our production team.

Kind regards,

on behalf of

Dr. Wolfgang Blenau

Academic Editor

PLOS One